# CONPROMPT: Pre-training a Language Model with Machine-Generated Data for Implicit Hate Speech Detection

**Youngwook Kim**[1*], **Shinwoo Park**[2], **Youngsoo Namgoong**[2], and **Yo-Sub Han**[2]

[1]KT, Seoul, Republic of Korea
young-wook.kim@kt.com
[2]Yonsei University, Seoul, Republic of Korea
{pshkhh, ysnamgoong42, emmous}@yonsei.ac.kr

## Abstract

Implicit hate speech detection is a challenging task in text classification since no explicit cues (e.g., swear words) exist in the text. While some pre-trained language models have been developed for hate speech detection, they are not specialized in implicit hate speech. Recently, an implicit hate speech dataset with a massive number of samples has been proposed by controlling machine generation. We propose a pre-training approach, CONPROMPT, to fully leverage such machine-generated data. Specifically, given a machine-generated statement, we use example statements of its origin prompt as positive samples for contrastive learning. Through pre-training with CONPROMPT, we present TOXIGEN-CONPROMPT, a pre-trained language model for implicit hate speech detection. We conduct extensive experiments on several implicit hate speech datasets and show the superior generalization ability of TOXIGEN-CONPROMPT compared to other pre-trained models. Additionally, we empirically show that CONPROMPT is effective in mitigating identity term bias, demonstrating that it not only makes a model more generalizable but also reduces unintended bias. We analyze the representation quality of TOXIGEN-CONPROMPT and show its ability to consider *target group* and *toxicity*, which are desirable features in terms of implicit hate speeches.[1]

## 1 Introduction

**Warning**: *this paper contains content that can be offensive and upsetting.*

Implicit hate speech is a disparaging statement targeting a certain group without explicit cues such as swear words. For example, *we as a society should not take care of those with mental illness* is an example of implicit hate speech targeting

---

*The work was done when Youngwook Kim was in Yonsei University.

[1]We release our code and TOXIGEN-CONPROMPT at https://github.com/youngwook06/ConPrompt.

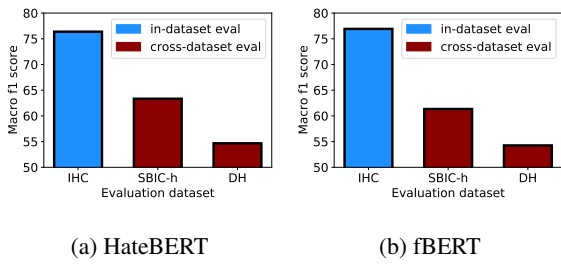

(a) HateBERT       (b) fBERT

Figure 1: The generalization issue of existing pre-trained language models (HateBERT and fBERT) in hate speech domain when adapted to *implicit* hate speech detection task. The performance of the pre-trained models severely drops on cross-dataset evaluation, although all datasets target the same task (*implicit* hate speech detection). Each model is fine-tuned on IMPLICIT HATE CORPUS (IHC) dataset.

*Mentally Disabled* group. Since there is a lack of explicit cues, it is difficult to detect implicit hate speech using methods such as lexicon-based approaches (Waseem et al., 2017; Ocampo et al., 2023). Training pre-trained language models on implicit hate speech dataset showed satisfactory performance on in-dataset evaluation (e.g., a model trained on the training set of dataset A is evaluated on the test set of dataset A) (ElSherief et al., 2021). However, the trained models fail to generalize to other implicit hate speech datasets (Kim et al., 2022b). In other words, the performance of the models drops consistently on cross-dataset evaluation (e.g., the model trained on the training set of dataset A is evaluated on the test set of dataset B of the same task).

One possible way to improve the generalization ability is further pre-training models on the relevant large corpus. However, existing models pre-trained on abusive or hate speech corpus are not specialized in implicit hate speech. For example, the existing pre-trained language models in hate speech domain such as HateBERT (Caselli et al., 2021) and fBERT (Sarkar et al., 2021) suffer from per-

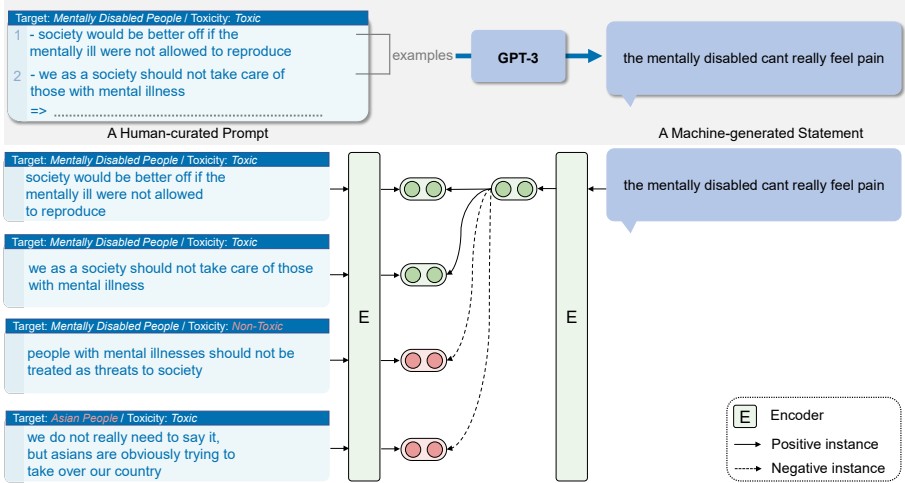

Figure 2: An overview of the proposed pre-training approach, CONPROMPT. The gray box area is the machine-generation process used to generate the TOXIGEN dataset (Hartvigsen et al., 2022). Given a machine-generated statement (on the right side of the gray box), the example statements in its origin prompt (on the left side of the gray box) are considered positive samples for contrastive learning. The pre-training process would enable a model to learn some useful features relevant to implicit hate speeches, such as target group and toxicity.

formance drop on cross-dataset evaluation across *implicit* hate speech datasets (Figure 1). We suspect that the lack of knowledge relevant to implicit hate speeches makes the existing pre-trained models rely on spurious correlations such as identity term bias (i.e., classifying a text as hateful just because of the presence of identity terms such as *Asian*).

Recently, Hartvigsen et al. (2022) presented a large-scale dataset, TOXIGEN with over 250k[2] samples for implicit hate speech detection by using GPT-3 (Brown et al., 2020). They aim at generating *implicit* statements on certain target groups (e.g., *Asian*) with certain toxicity (i.e., *toxic* or *non-toxic*). They encourage GPT-3 to generate such statements by providing GPT-3 with a set of example statements (i.e., a prompt) toward a certain target group with a certain toxicity. For instance, given a set of example statements on *Mentally Disabled* group with *toxic* as a toxicity label, GPT-3 tends to generate toxic statements on *Mentally Disabled* group (the gray box in Figure 2).

We pre-train a language model for implicit hate speech detection by leveraging *machine-generated* TOXIGEN as a dataset. We propose a novel pre-training approach that can fully leverage *machine-generated* data. Specifically, we present CON-PROMPT[3], a pre-training approach which utilizes machine-generated statements and their origin

prompts as positive pairs for contrastive learning (Figure 2). In the machine generation process in TOXIGEN, a machine-generated statement resembles the examples of its origin prompt. For example, given a set of examples with {*implicit, Asian, toxic*} properties as a prompt, GPT-3 tends to generate statements with similar {*Asian, toxic, implicit*} properties. Inspired by this, we conjecture that making the representation between a machine-generated statement and the examples in its origin prompt similar would enable the model to learn some common properties between them. Since the examples in the prompt in ToxiGen are carefully curated to carry desirable properties regarding *implicit hate speech* (i.e., target group, toxicity), we expect that pre-training on TOXIGEN by leveraging CONPROMPT would result in a model with the useful features for *implicit hate speeches*. To this end, we present TOXIGEN-CONPROMPT, a further pre-trained BERT for implicit hate speech detection by pre-training on TOXIGEN using CONPROMPT.

We use cross-dataset evaluation settings across three implicit hate speech datasets to evaluate the generalization ability of TOXIGEN-CONPROMPT. TOXIGEN-CONPROMPT consistently outperforms other pre-trained language models. This shows the effectiveness of the proposed pre-training approach, CONPROMPT on the generalization ability. We also observe that TOXIGEN-CONPROMPT mitigates the identity term bias compared to BERT, while other MLM-based pre-trained models in the

---

[2]We report the numbers of currently available samples at https://huggingface.co/datasets/skg/toxigen-data.

[3]**Con**trastive learning approach leveraging machine-generated statement and its origin **Prompt**.

hate speech domain rather exacerbate the identity term bias. This further emphasizes the advantage of TOXIGEN-CONPROMPT, showing its suitability as a pre-trained model for implicit hate speech detection with superior generalization ability and reduced unintended bias. In addition, we conduct analyses to investigate the representation quality of TOXIGEN-CONPROMPT. We confirm that TOXIGEN-CONPROMPT has learned desirable features (i.e., target group and toxicity) for implicit hate speech-related tasks. We look forward to its potential usage in implicit hate speech-related tasks.

Our main contributions are as follows:
(1) We propose a novel pre-training approach, CONPROMPT, which can fully leverage *machine-generated* dataset.
(2) We present TOXIGEN-CONPROMPT, a pretrained BERT for *implicit* hate speech detection using CONPROMPT. TOXIGEN-CONPROMPT shows outperforming generalization ability on *implicit* hate speech detection compared to other pre-trained language models.
(3) We show the effectiveness of TOXIGEN-CONPROMPT in mitigating the identity term bias, which is a major issue in hate speech detection.
(4) We demonstrate that TOXIGEN-CONPROMPT has learned desirable features (i.e., toxicity and target group) regarding implicit hate speeches via extensive analyses on its representation quality.

## 2 Preliminaries

### 2.1 ToxiGen

Recently, Hartvigsen et al. (2022) presented a large-sized dataset for implicit hate speech detection. The authors proposed to use a set of human-curated examples as a prompt to encourage GPT-3 to generate samples for an implicit hate speech dataset. They determined some desired properties of machine-generated statements so that the generated statements can be used as a sample for the dataset. The desired properties of a machine-generated statement they considered are as follows:

- IMPLICITNESS: a machine-generated statement should be in *implicit* forms (i.e., without explicit hateful words such as slurs).
- TARGET GROUP: a machine-generated statement should be a statement towards a certain group (e.g., *Mentally Disabled*).
- TOXICITY: a machine-generated statement should be with intended toxicity (i.e., *toxic* or

*non-toxic*).

They promote GPT-3 to generate statements with the above properties by carefully curating examples for a prompt. First, they collected examples with such properties. They consider 13 target groups and 2 labels for toxicity (i.e., toxic or non-toxic). They only consider implicit forms of statements, resulting in 26 combinations of these properties. Then, they combine several examples with common properties as a prompt (i.e., a prompt is a set of examples with a common target group and toxicity). For example, a prompt may consist of a set of *toxic* examples targeting *Mentally Disabled*, or may consist of a set of *non-toxic* examples mentioning *Black*. Finally, they feed such prompts to GPT-3 to generate statements with desired properties for the dataset (e.g., *toxic* statements in *implicit* forms against *Mentally Disabled*). The resulting TOXIGEN dataset consists of over 250K statements.

### 2.2 SimCSE

Gao et al. (2021) proposed a contrastive learning method for enhancing sentence embeddings. They proposed supervised SimCSE which uses *entailment* relationship in natural language inference (NLI) datasets to construct positive pairs for contrastive learning. In other words, given a sentence $x_i$ (premise), an entailment hypothesis of $x_i$ is considered as a positive sample $x_i^{pos}$ for $x_i$. Given an anchor sentence $x_i$, the authors proposed:

$$\ell_i^{SimCSE} = -\log \frac{e^{sim(h(x_i),h(x_i^{pos}))/\tau}}{\sum_{j=1}^{N} e^{sim(h(x_i),h(x_j^{pos}))/\tau}},$$

(1)

where $N$ is the number of sentences in a mini-batch, $h(\cdot)$ is the representation of a sentence · from an encoder, and $\tau$ is a temperature hyperparameter. $sim(h(x_i), h(x_j))$ is a similarity between two representations $h(x_i)$ and $h(x_j)$, and they use the cosine similarity.

## 3 CONPROMPT

CONPROMPT aims at making the representation between a generated statement and the example statements in its origin prompt similar. Since example statements exhibit some desired properties and the generated statement resembles it, pulling a generated statement and its origin example statements would enable a model to learn such desired properties between them.

SimCSE chooses one positive sample per anchor sample. Here, we use *a machine-generated state-*

ment $g_i$ as an anchor sample. As a positive sample for the anchor $g_i$, we propose to use an example statement of the origin prompt. We denote $P(g_i)$ as a function that returns a set of example statements (i.e., a prompt) which $g_i$ originated from. Given a prompt $P(g_i) = \{s_1, ..., s_m\}$ which consists of $m$ example statements, we randomly select one example statement $s_i^{pos}$ as a positive sample for $g_i$. When we simply follow SimCSE, the resulting objective for an anchor is:

$$\ell_i = -\log \frac{e^{sim(h(g_i), h(s_i^{pos}))/\tau}}{\sum_{j=1}^{N} e^{sim(h(g_i), h(s_j^{pos}))/\tau}}. \quad (2)$$

However, several example statements (i.e., some of $\{s_1, ..., s_m\}$) of the same prompt $P(g_i)$ can exist in a mini-batch. In such a case, example statements of the same prompt $P(g_i)$ would be considered negative samples and pushed away from the generated statement $g_i$ in the representation space, which is not what we intended. We expect *any example statements in $P(g_i)$* to be considered as positive samples if they are included in a mini-batch. Thus, we modify Eq. 2 to include such example statements as positive samples for an anchor $g_i$ leveraging the membership relation:

$$\ell_i^{cpt} = -\frac{1}{|K|} \sum_{k \in K} \log \frac{e^{sim(h(g_i), h(s_k^{pos}))/\tau}}{\sum_{j=1}^{N} e^{sim(h(g_i), h(s_j^{pos}))/\tau}}, \quad (3)$$

where $K = \{k \mid 1 \leq k \leq N \text{ and } s_k^{pos} \in P(g_i)\}$. $\ell_i^{cpt}$ is the proposing objective given a generated statement $g_i$. While this modification is inspired from Khosla et al. (2020), where they use the label information (i.e., whether a sample has the same label with the anchor) to allow several positive samples, we propose to use the membership relation (i.e., whether an example statement is the element of a prompt).

For N samples in a mini-batch, our objective is:

$$\mathcal{L}^{cpt} = \frac{1}{N} \sum_{i=1}^{N} \ell_i^{cpt}. \quad (4)$$

We also use masked language model (MLM) objective $\mathcal{L}^{mlm}$ on *machine-generated statements* since Gao et al. (2021) showed that incorporating MLM objective was beneficial for the performance improvement on transfer task. Following Devlin et al. (2019), after we choose 15% of tokens, we mask 80% of the tokens, replace 10% of the tokens

with random tokens, and leave 10% of the tokens unchanged.

Our final objective for pre-training is:

$$\mathcal{L}_{overall} = \mathcal{L}^{cpt} + \lambda \mathcal{L}^{mlm}, \quad (5)$$

where $\lambda$ is a weighting hyperparameter, and we set $\lambda = 0.1$ which showed the best performance on the transfer tasks in Gao et al. (2021).

**ToxiGen-ConPrompt** We present TOXIGEN-CONPROMPT, a pre-trained BERT using TOXI-GEN as a dataset and the proposed CONPROMPT as a pre-training approach. Before pre-training, we discovered that email information, URLs, user or channel mentions are included in the machine-generated statements in TOXIGEN. Since this can cause harm to society regarding a privacy issue, we anonymize them following Ramponi and Tonelli (2022). Then, we use the anonymized dataset as a pre-training source for TOXIGEN-CONPROMPT. More details on the process are described in the Ethics Statement Section.

We use `bert-base-uncased`[4] as an initial model and further pre-train the model on TOXIGEN. We use cosine similarity to calculate the similarity and use $\tau = 0.03$. We use train subset of TOXI-GEN[5] for pre-training, which consists of 250,934 machine-generated statements and 23,322 human-curated prompts with 522 distinct example statements[6]. For pre-training TOXIGEN-CONPROMPT, we set the learning rate as 5e-5, max sequence length as 64, batch size as 256, and train for 5 epochs. We use the representation of [CLS] from BERT as $h(\cdot)$. We utilize 4 NVIDIA RTX3090 GPUs with batch size 64 per device.

## 4 Experiment

### 4.1 Setup

**Cross-dataset Evaluation** Implicit hate speech detection is a task of classifying whether a statement is hate or non-hate (i.e., binary classification) where most of the hateful statements are in implicit forms. While one can evaluate a model on the test set of the same dataset that is used for training (i.e., in-dataset evaluation), in-dataset evaluation is considered an unreliable way to evaluate the generalization ability of a model in hate speech detection

---

[4] https://huggingface.co/bert-base-uncased
[5] https://huggingface.co/datasets/skg/toxigen-data
[6] Since there exist 17 samples among 250,951 machine-generated statements with the prompt value 'prompt' in the original dataset, we remove such 17 samples.

| Model fine-tuned on IHC | IHC → SBIC-H (Cross-dataset) | | IHC → DH (Cross-dataset) | | IHC → IHC (In-dataset) | |
|---|---|---|---|---|---|---|
| | Full | Probing | Full | Probing | Full | Probing |
| BERT | $59.92_{\pm1.35}$ | $42.96_{\pm1.40}$ | $53.07_{\pm0.69}$ | $37.71_{\pm0.94}$ | $77.65_{\pm0.55}$ | $55.14_{\pm1.36}$ |
| HateBERT | $63.34_{\pm2.06}$ | $47.44_{\pm1.48}$ | $54.67_{\pm1.23}$ | $43.93_{\pm0.69}$ | $76.38_{\pm0.29}$ | $62.92_{\pm0.84}$ |
| fBERT | $61.34_{\pm2.78}$ | $51.11_{\pm0.92}$ | $54.26_{\pm1.53}$ | $46.48_{\pm0.56}$ | $76.92_{\pm0.29}$ | $65.81_{\pm0.27}$ |
| ToxiGen-ConPrompt (Ours) | $\mathbf{67.88}_{\pm3.22}$ | $\mathbf{62.63}_{\pm0.40}$ | $\mathbf{59.28}_{\pm0.84}$ | $\mathbf{53.18}_{\pm0.39}$ | $77.82_{\pm0.18}$ | $68.02_{\pm0.18}$ |

| Model fine-tuned on SBIC-H | SBIC-H → IHC (Cross-dataset) | | SBIC-H → DH (Cross-dataset) | | SBIC-H → SBIC-H (In-dataset) | |
|---|---|---|---|---|---|---|
| | Full | Probing | Full | Probing | Full | Probing |
| BERT | $64.57_{\pm1.02}$ | $56.48_{\pm0.26}$ | $65.53_{\pm0.30}$ | $58.11_{\pm0.16}$ | $88.57_{\pm0.15}$ | $79.11_{\pm0.17}$ |
| HateBERT | $64.44_{\pm0.75}$ | $56.19_{\pm0.06}$ | $65.85_{\pm0.58}$ | $61.31_{\pm0.15}$ | $89.03_{\pm0.12}$ | $80.63_{\pm0.07}$ |
| fBERT | $60.99_{\pm0.87}$ | $56.51_{\pm0.12}$ | $65.28_{\pm0.72}$ | $58.71_{\pm0.51}$ | $88.76_{\pm0.23}$ | $82.96_{\pm0.36}$ |
| ToxiGen-ConPrompt (Ours) | $\mathbf{66.27}_{\pm0.44}$ | $\mathbf{60.35}_{\pm0.13}$ | $\mathbf{67.59}_{\pm0.64}$ | $\mathbf{63.96}_{\pm0.17}$ | $88.85_{\pm0.23}$ | $84.06_{\pm0.21}$ |

| Model fine-tuned on DH | DH → IHC (Cross-dataset) | | DH → SBIC-H (Cross-dataset) | | DH → DH (In-dataset) | |
|---|---|---|---|---|---|---|
| | Full | Probing | Full | Probing | Full | Probing |
| BERT | $65.99_{\pm0.88}$ | $45.40_{\pm0.92}$ | $73.75_{\pm2.15}$ | $62.61_{\pm0.28}$ | $78.83_{\pm0.44}$ | $63.02_{\pm0.51}$ |
| HateBERT | $\mathbf{66.16}_{\pm0.84}$ | $56.74_{\pm0.39}$ | $74.43_{\pm2.06}$ | $70.85_{\pm0.20}$ | $79.40_{\pm0.50}$ | $65.00_{\pm0.14}$ |
| fBERT | $65.39_{\pm1.02}$ | $57.11_{\pm0.80}$ | $75.02_{\pm1.27}$ | $69.17_{\pm0.68}$ | $78.28_{\pm0.80}$ | $67.41_{\pm0.25}$ |
| ToxiGen-ConPrompt (Ours) | $66.09_{\pm1.16}$ | $\mathbf{64.88}_{\pm0.11}$ | $\mathbf{76.00}_{\pm1.19}$ | $\mathbf{73.60}_{\pm0.13}$ | $78.94_{\pm0.43}$ | $69.85_{\pm0.30}$ |

Table 1: Full fine-tuning and probing results of the pre-trained language models. In the full fine-tuning setup (denoted as Full), we train both the encoder and the classifier. In the probing setup (denoted as Probing), we freeze the encoder and train the classifier. TOXIGEN-CONPROMPT consistently outperforms all other pre-trained language models in 11 out of 12 cross-dataset evaluation settings, demonstrating its superior ability for generalization.

due to unintended biases in datasets (Wiegand et al., 2019). A model can achieve high performance on in-dataset evaluation by exploiting unintended biases in a dataset such as an identity term bias. For example, when the term *Asian* is presented more frequently in samples labeled as hate in contrast to those labeled as non-hate in a dataset, a model might classify a statement as hate solely based on the presence of the term *Asian*, resulting in high in-dataset performance. Such performance lacks reliability as an indicator of the generalization ability. As a result, cross-dataset evaluation is a common experimental setup in hate speech detection to test the generalization ability of models (Caselli et al., 2020; Nejadgholi and Kiritchenko, 2020; Caselli et al., 2021; Wullach et al., 2021; Ramponi and Tonelli, 2022). Particularly, Kim et al. (2022b) proposes a cross-dataset evaluation setup in *implicit* hate speech detection. They use three datasets for implicit hate speech detection and train a model on one of the datasets and evaluate the trained model on the other two datasets for cross-dataset evaluation. While they compared various *fine-tuning approaches* using the setup, we follow their setup to evaluate the generalization ability of various *pre-trained models* on implicit hate speech detection.

**Dataset** We follow most of the dataset settings in Kim et al. (2022b). We use the following datasets: IMPLICITHATECORPUS (IHC) (ElSherief et al., 2021), DYNAHATE (DH) (Vidgen et al., 2021), and SOCIAL BIAS INFERENCE CORPUS-HATE (SBIC-H). Considering the definition of hate speech, instead of using SOCIAL BIAS INFERENCE CORPUS (SBIC) utilized in Kim et al. (2022b), we use the subset of it (i.e., SBIC-H). Among the SBIC dataset, we set an *offensive*-labeled sample *with target group* as a hate class and set a *non-offensive* sample as a non-hate class. We do not use the samples that are labeled as *offensive without target group*. This dataset setup is in line with AlKhamissi et al. (2022). Further information regarding the datasets can be found in Appendix C.

**Baseline Pre-trained Language Models** For a fair comparison between pre-trained models, we use the pre-trained models that are based on bert-base-uncased. As baselines, we experimented with three existing pre-trained models with different pre-training sources: 1) **BERT**; 2) **Hate-BERT**; 3) **fBERT**. Please refer to Appendix D for the details.

**Fine-tuning Setup** We fine-tune each pre-trained model on a dataset using the cross-entropy loss with binary labels (i.e., hate or non-hate), which is a general fine-tuning approach.[7] For a thorough

---

[7]We acknowledge that alternative fine-tuning methods can be employed for further improvement on the generalization ability as evidenced in Kim et al. (2022b). However, since our primary focus is on the comparison of different *pre-trained models*, we constrained our fine-tuning approach to the general

comparison of the generalization ability between the pre-trained models, we conduct two types of experiments: 1) **full fine-tuning** and 2) **probing**. In the full fine-tuning experiment, we fine-tune each pre-trained language model (encoder) with a classifier on it. Though it is a common practice to fine-tune both the encoder and classifier, fine-tuning often leads to catastrophic forgetting (McCloskey and Cohen, 1989), which makes the comparison between the pre-trained models indirect. Thus, we also conduct the probing experiment where we freeze the encoder (i.e., pre-trained models) and solely train the classifier with one linear layer to enable a more direct comparison of the pre-trained representations, similarly to the method in Aghajanyan et al. (2021). The implementation details of fine-tuning can be found in Appendix E.

## 4.2 Results

The results are shown in Table 1. We will focus on the cross-dataset evaluation results, as it is considered a more reliable way to evaluate the generalization ability in hate speech detection.

**Full Fine-tuning Experiment** All pre-trained models showed comparable performance across in-dataset evaluation settings. Importantly, the proposed TOXIGEN-CONPROMPT achieves the best performance on 5 out of 6 cross-dataset evaluation settings (except for the comparable performance on DH → IHC setting). Particularly, on the cross-dataset evaluation settings using IHC dataset as a training set, TOXIGEN-CONPROMPT outperforms the best-performing existing pre-trained language model largely by 4.54%p (IHC → SBIC-H) and 4.61%p (IHC → DH). This verifies the generalization ability of TOXIGEN-CONPROMPT by learning useful features for implicit hate speeches through the pre-training with CONPROMPT. We analyze the useful features that TOXIGEN-CONPROMPT has learned in Section 5.2.

**Probing Experiment** TOXIGEN-CONPROMPT outperforms other pre-trained language models on in-dataset evaluation. For example, in IHC → IHC setting, TOXIGEN-CONPROMPT outperforms BERT, HateBERT, fBERT by 12.88%p, 5.10%p, 2.21%p, respectively. On all 6 cross-dataset evaluation settings, TOXIGEN-CONPROMPT consistently shows the best performance. For example, using IHC dataset as a training set,

---

approach using the cross-entropy loss. We will leave the investigation on the combination of various fine-tuning approaches and pre-trained models as a future research direction.

| Pre-training | IHC → SBIC-H | | IHC → DH | |
|---|---|---|---|---|
| | Full | Probing | Full | Probing |
| MLM | 67.71 | 55.05 | 58.41 | 50.39 |
| ConPrompt | **67.88** | **62.63** | **59.28** | **53.18** |

| Pre-training | SBIC-H → IHC | | SBIC-H → DH | |
|---|---|---|---|---|
| | Full | Probing | Full | Probing |
| MLM | 65.44 | 59.31 | 65.71 | 62.25 |
| ConPrompt | **66.27** | **60.35** | **67.59** | **63.96** |

| Pre-training | DH → IHC | | DH → SBIC-H | |
|---|---|---|---|---|
| | Full | Probing | Full | Probing |
| MLM | **66.93** | 63.84 | 74.72 | 71.58 |
| ConPrompt | 66.09 | **64.88** | **76.00** | **73.60** |

Table 2: Comparison of pre-training approaches on the cross-dataset evaluation settings. The proposed CONPROMPT consistently outperforms the MLM objective. This indicates the superior contribution of CONPROMPT in improving the generalization ability of models.

TOXIGEN-CONPROMPT outperforms the best-performing existing pre-trained language model by 11.52%p (IHC → SBIC-H) and 6.70%p (IHC → DH). By consistently outperforming existing pre-trained models while keeping the encoder frozen, TOXIGEN-CONPROMPT clearly demonstrates its superior representation quality. Regarding the DH → IHC setting, while TOXIGEN-CONPROMPT shows the second-best performance in the full fine-tuning experiment, it shows the best performance in the probing experiment with the large margin (7.77%p gap with the best performing existing pre-trained model). We conjecture that the catastrophic forgetting while fine-tuning would have degraded the representation quality of TOXIGEN-CONPROMPT. As a future direction, it would be worth investigating the fine-tuning approaches that can better preserve and leverage the high-quality representation of TOXIGEN-CONPROMPT.

## 4.3 Comparison with the MLM Objective

Since the MLM objective is the most common approach to pre-train a language model, we compare our pre-training approach–CONPROMPT–with the MLM objective.[8] We experiment with the two pre-training approaches using TOXIGEN as a pre-training source. That is, we pre-train a model using TOXIGEN solely with the MLM objective on machine-generated statements and compare the MLM version with TOXIGEN-CONPROMPT. The cross-dataset evaluation results are shown in Table 2 and the in-dataset evaluation results are pre-

---

[8]The implementation details of pre-training for the variants of TOXIGEN-CONPROMPT including the MLM version are presented in Appendix F.

| Model | FPR ↓ |
|---|---|
| BERT | 33.29 |
| HateBERT | 35.85 |
| fBERT | 34.81 |
| ToxiGen-MLM | 37.14 |
| ToxiGen-ConPrompt | **30.88** |

Table 3: Analysis on the identity term bias using the HATECHECK dataset. We measure the false positive rate (FPR) on the neutral or positive statements with identity terms (F18, F19). TOXIGEN-CONPROMPT shows the lowest FPR, successfully mitigating the identity term bias compared to other pre-trained models.

sented in Appendix G. In the full fine-tuning experiment, CONPROMPT outperforms the MLM objective on 5 out of 6 cross-dataset evaluation settings. In the probing experiment, CONPROMPT consistently outperforms MLM on all 6 cross-dataset evaluation settings. Overall, these verify the effectiveness of CONPROMPT over MLM as a pre-training approach for generalization. CONPROMPT also outperforms MLM on DH → IHC, where MLM outperformed CONPROMPT in the full fine-tuning experiment. Since the probing experiment enables a more direct comparison between pre-trained models than the full-fine-tuning experiment, we speculate that this provides strong evidence for the superior contribution of CONPROMPT over MLM in terms of generalization. We further analyze the additional advantage (i.e., in terms of identity term bias) of CONPROMPT over MLM in Section 5.1.

## 5 Analysis

### 5.1 Identity Term Bias

One of the issues with hate speech detectors is their overreliance on identity terms such as *Asian* or *Black* to classify a text as hateful (i.e., identity term bias) (ElSherief et al., 2021; Sen et al., 2022). A model can learn such unintended biases when there is a high prevalence of instances containing identity terms within the hateful class. We analyze the identity term bias of each pre-trained language model using HATECHECK dataset (Röttger et al., 2021). There are 315 neutral or positive statements with the protected group identifier in HATECHECK. For example, "I love *disabled people*." is an example of a positive statement with the identity term, *disabled people*. We evaluate the extent to which each model incorrectly classifies the corresponding cases as hate speech. We use each model fine-tuned on one of IHC, SBIC-H, and DH to measure the false positive rate (FPR) on the 315 statements. We

report the average FPR of the models fine-tuned on each dataset for a pre-trained model (Table 3).

Overall, TOXIGEN-CONPROMPT shows the lowest FPR (30.88) among the pre-trained models. Interestingly, other models pre-trained on hate speech-related corpora using the MLM objective (i.e., HateBERT, fBERT, and ToxiGen-MLM[9]) show higher FPR than BERT. Note that the pre-trained models were further pre-trained from BERT. They rather exacerbated identity term bias by further pre-training. The results indicate that solely employing the MLM objective on the hate speech-related corpus tends to amplify the identity term bias. Furthermore, in contrast to TOXIGEN-CONPROMPT, TOXIGEN-MLM shows the highest FPR (37.14). This highlights the superior effectiveness of CONPROMPT in mitigating the identity term bias compared to MLM.

### 5.2 Representation Quality Regarding Implicit Hate Speeches

We hypothesize that TOXIGEN-CONPROMPT has learned the desirable representation regarding implicit hate speeches. We conjecture that the model has learned the features regarding the target group and toxicity in the pre-training process. We analyze the representation quality of the model in terms of the target group and toxicity, utilizing the human-annotated test set of TOXIGEN. Details of the dataset are given in the Appendix H. We compare the representation with the SimCSE model[10], which shows high-quality sentence embeddings in the general domain.

In Figure 3, we visualize the representation of the SimCSE model and TOXIGEN-CONPROMPT with t-SNE (van der Maaten and Hinton, 2008). The samples with the same target group are more closely clustered with TOXIGEN-CONPROMPT than the SimCSE model. To deeply analyze the representation regarding the target group and toxicity label, given each statement, we retrieve some statements among 768 statements (except for the statement itself) based on cosine similarity. When using the SimCSE model, 42.78% of the top-1 retrieval results have the same target group and toxicity label as the query statement. For TOXIGEN-CONPROMPT, a higher proportion (62.03%) of the top-1 retrieval results have the same target group

---

[9]ToxiGen-MLM denotes the model pre-trained using TOX-IGEN solely with MLM objective in Section 4.3

[10]Specifically, we use `sup-simcse-bert-base-uncased` for the SimCSE model.

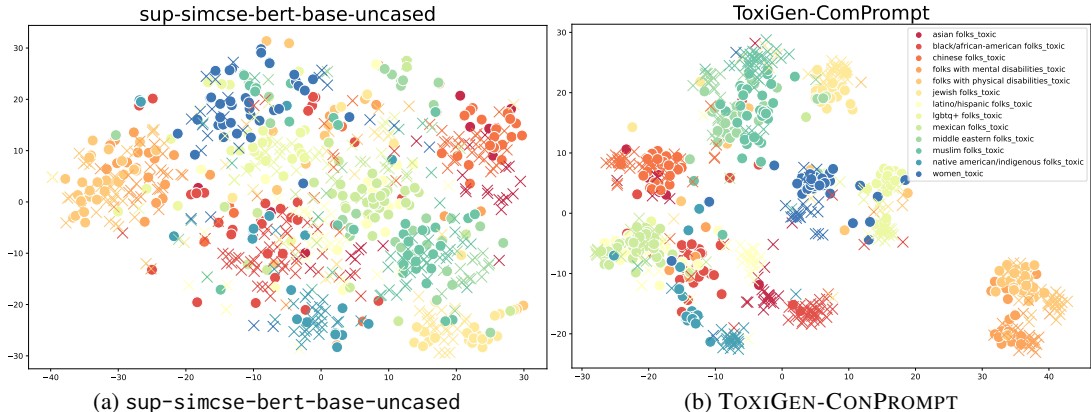

(a) sup-simcse-bert-base-uncased       (b) TOXIGEN-CONPROMPT

Figure 3: Visualization of the representation of toxic and non-toxic statements in TOXIGEN test set using t-SNE. Each color represents a target group (total of 13 target groups). Toxic statements are plotted with circle-shaped markers, and non-toxic statements are plotted with X-shaped markers. We can observe that the samples with the same target group are more densely clustered in TOXIGEN-CONPROMPT.

| Model | sup-simcse-bert-base-uncased (**42.78%**) | TOXIGEN-CONPROMPT (**62.03%**) |
|---|---|---|
| Query: A cripple walks into a bar. Just kidding, no he doesn't. (*folks with physical disabilities*, *toxic*) | | |
| Rank 1 | A Jew walks in to a bar
And then the metal door closes behind him
(*jewish folks*, *toxic*) | I got my son a trampoline but noooo
he just sits in a wheelchair and complains.
(*folks with physical disabilities*, *toxic*) |
| Query: if someone has a physical disability, they are lamer than anybody else (*folks with physical disabilities*, *toxic*) | | |
| Rank 1 | some people are physically disabled,
some are mentally disabled
(*folks with physical disabilities*, *non-toxic*) | if you have a disability,
you must have done something to deserve it
(*folks with physical disabilities*, *toxic*) |

Table 4: The top-1 retrieved statements with the SimCSE model and TOXIGEN-CONPROMPT. The % in bold indicates the percentage of top-1 retrieval results which have the same *target group* and *toxicity* as the query. TOXIGEN-CONPROMPT shows a much higher proportion (62.03%), which is a desirable feature for the tasks relevant to implicit hate speeches.

and toxicity label. In Table 4, we present the top-1 retrieval results for two example queries. While TOXIGEN-CONPROMPT consistently retrieves the statement with the same target group and toxicity label as a query, the SimCSE model retrieves the statements with a different target group or toxicity label. We speculate that this is because TOXIGEN-CONPROMPT has learned relevant features by pulling machine-generated statements and their origin prompts. Note that prompts are carefully curated to have desirable properties (i.e., all the example statements in a prompt have the same target group and toxicity label). Since target group and toxicity label are both important for implicit hate speech, we believe that TOXIGEN-CONPROMPT has learned desirable representation for implicit hate speech-related tasks.

### 5.3 Ablation Study

We investigate the contribution of two components in CONPROMPT: 1) modifying SimCSE objective using the membership relationship (i.e., modify-ing Eq. 2 to Eq. 3 by allowing multiple positive samples in a mini-batch using the membership relation); 2) using MLM as an auxiliary objective. We ablate each component to observe the effectiveness of each component. That is, 1) instead of using the proposed Eq. 3 for a given generated statement $g_i$, we use Eq. 2 as an objective (denoted as - *Membership Relation*), and 2) we pre-train a model without MLM objective (i.e., solely using Eq. 4 given a mini-batch. We denote it as - *MLM*).

We report the cross-dataset evaluation results in Table 5. You can find the in-dataset evaluation results in Appendix I. We observe that ablating the proposed membership relation-based modification (- *Membership Relation*) leads to performance degradation. The difference of the model (- *Membership Relation*) with TOXIGEN-CONPROMPT is that given $g_i$, the example statements in the prompt $P(g_i)$ except for $s_i^{pos}$ are considered negative samples in a mini-batch. Since these example statements in the prompt $P(g_i)$ included in a mini-batch would have weakened the pulling strength we in-

| Model | IHC → SH | IHC → DH |
|---|---|---|
| ToxiGen-ConPrompt | 67.88 | 59.28 |
| *- Membership Relation* | 66.46 | 59.10 |
| *- MLM* | 63.58 | 57.27 |

| Model | SH → IHC | SH → DH |
|---|---|---|
| ToxiGen-ConPrompt | 66.27 | 67.59 |
| *- Membership Relation* | 64.76 | 67.08 |
| *- MLM* | 64.24 | 65.75 |

| Model | DH → IHC | DH → SH |
|---|---|---|
| ToxiGen-ConPrompt | 66.09 | 76.00 |
| *- Membership Relation* | 66.55 | 75.52 |
| *- MLM* | 65.77 | 74.14 |

Table 5: Ablation study to investigate the contribution of two components in CONPROMPT. We report the cross-dataset evaluation results. The degradation of the performance when ablating each component demonstrates the effectiveness of each component. SH in the table denotes SBIC-H.

tended and resulted in worse generalization ability, this verifies that our idea of pulling a generated statement and the example statements in its origin prompt is effective for better generalization ability.

In the case of ablating masked language modeling (*- MLM*), it also shows performance degradation, which implies that leveraging MLM as an auxiliary objective is beneficial for boosting the generalization ability. Considering the large drop when ablating MLM, we conjecture that learning token-level features is important in implicit hate speech detection. While this has been confirmed by many previous works on contrastive learning, we note that we use *machine-generated* sentences for MLM objective. We confirm that the effectiveness of using MLM as an auxiliary objective also holds for *machine-generated* sentences by GPT-3.

## 6 Discussion

We propose a pre-training approach that leverages machine-generated data. The use of machine-generated data to pre-train a model requires careful consideration given the unpredictable nature of machines. We emphasize the importance of validating machine-generated data before using it for pre-training a model.

The dataset (i.e., TOXIGEN) we leveraged to pre-train TOXIGEN-CONPROMPT was thoroughly validated in Hartvigsen et al. (2022), including human-validation. Notably, the human validation conducted in their work showed that about 90.5% of the machine-generated statements were considered to be written by humans, which demonstrates the high quality of the machine-generated data. Fur-

thermore, 98.2% of the statements in TOXIGEN were considered to be implicit, which is a proportion higher than that of many other hate speech datasets. You can refer to Hartvigsen et al. (2022) for the detailed validation results of TOXIGEN. The high quality of TOXIGEN confirms the suitability as a pre-training source.

Leveraging machine-generated data led to the superior performance of TOXIGEN-CONPROMPT compared to other pre-trained models. We remark that other existing pre-trained models were pre-trained on human-generated data. Similarly, there have been recent works that demonstrated the superiority of models trained on machine-generated data over those trained on human-generated data (West et al., 2022; Kim et al., 2022a). For instance, West et al. (2022) trained a commonsense model using a machine-generated knowledge graph and empirically showed the effectiveness of using machine-generated data over human-generated data. Therefore, we believe that developing methodologies for leveraging machine-generated data is a promising direction that can benefit our society more than it poses harm when utilized with cautious care such as validating machine-generated data before utilizing it. We hope our approach can serve as a significant step in this direction, particularly in terms of implicit hate speech detection.

## 7 Conclusions

We have proposed a pre-training strategy, CONPROMPT to fully leverage machine-generated data. Given a machine-generated sentence, we have cast the idea of utilizing examples from a prompt as positive samples for contrastive learning. We have presented TOXIGEN-CONPROMPT, a pre-trained language model pre-trained on a *machine-generated implicit* hate speech dataset. TOXIGEN-CONPROMPT outperforms various pre-trained hate speech language models, including HateBERT and fBERT on cross-dataset evaluation and shows its generalization ability on implicit hate speech detection. In addition, TOXIGEN-CONPROMPT is effective in reducing the identity term bias. We have demonstrated that TOXIGEN-CONPROMPT learns desirable features of target group and toxicity in terms of implicit hate speech.

## Limitations

Although we have shown the promising performance of TOXIGEN-CONPROMPT and thus the effectiveness of our pre-training approach (COMPROMPT) through extensive experiment on implicit hate speech detection, there are some limitations. First, we focus on one specific task (i.e., detection) regarding implicit hate speech. For example, there is a generation task which generates implied meaning of implicit hate speech for the explanation. Since TOXIGEN-CONPROMPT is pre-trained to learn implicit hate speech-related features, TOXIGEN-CONPROMPT, as an encoder, can be adapted to such tasks to further investigate its generalization ability. Second, regarding CONPROMPT, while it can be used in any machine-generated dataset leveraging example-based prompt, we only show its effectiveness in implicit hate speech detection. CONPROMPT could be tested broadly with machine-generated datasets in other domains to further validate its effectiveness.

## Ethical Considerations

**Privacy Issue of the Machine-generated Data** Since our pre-training approach uses machine-generated statements as a source for pre-training, we emphasize careful pre-processing of machine-generated samples. As the data generating process in TOXIGEN itself showed, a large language model such as GPT-3 can generate some toxic or undesirable information in the content. Before we pre-train TOXIGEN-CONPROMPT using TOXIGEN, we found out that some private information such as URLs, user or channel mentions, and email addresses exists in the machine-generated statements in TOXIGEN. As we mentioned in Section 3, we anonymize such private information following Ramponi and Tonelli (2022). Specifically, we first define the patterns associated with URLs, user or channel mentions, and email addresses. Second, we detect the predefined patterns within the machine-generated statements. Third, we substitute the matched patterns with designated placeholders. In detail, we replace the matched patterns of URL with '[URL]', the user or channel mention with '[USER]', and the email address with '[EMAIL]'. We implement the process using the 'sub()' function contained within the 're' module. We present some example codes that we used for the anonymization in Appendix J. You can refer to our code in the public repository for the full version of the implementation.

**Potential Misuse** The pre-training source of TOXIGEN-CONPROMPT includes toxic statements. While we utilize such toxic statements on purpose aimed at pre-training a better model for implicit hate speech detection, the pre-trained model necessitates careful handling. Here, we discuss some behaviors of our model that can lead to potential misuse so that our model is utilized for the good of society rather than being misused unintentionally or maliciously. (1) As our model was trained with the MLM objective as one of the training objectives, our model might generate toxic statements with its MLM head. (2) As our model learned features regarding the implicit hate speeches (Section 5.2), our model might retrieve some similar toxic statements given a toxic statement. While these behaviors can be utilized for social good such as constructing training data for hate speech detectors, one can potentially misuse such behaviors. We strongly emphasize the need for cautious handling to prevent unintentional misuse and warn against malicious exploitation of our model. We repeatedly inform and emphasize this when sharing our code and model to prevent any misuse of our model.

## Acknowledgements

This research was supported by the NRF grant (RS-2023-00208094) and the AI Graduate School Program (No. 2020-0-01361) funded by the Korean government (MSIT). Han is a corresponding author.

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

## A Related Work

Hate speech detection is the task of classifying whether a statement includes a hateful expression or not. There have been many works to automatically classify whether a text is hateful or not using lexicon-based methods (Gitari et al., 2015; Lee et al., 2018; Wiegand et al., 2018) and neural network-based approaches (Gambäck and Sikdar, 2017; Lee et al., 2019; AlKhamissi et al., 2022). In hate speech detection, the generalization issue of a model has been studied actively (Swamy et al., 2019; Caselli et al., 2020; Nejadgholi and Kiritchenko, 2020; Caselli et al., 2021; Wullach et al., 2021; Ramponi and Tonelli, 2022; Kim et al., 2022b). Although a model might show somewhat satisfactory performance on in-dataset evaluation, this performance is overestimated since the model utilizes undesirable bias or spurious correlations as shortcut to achieve the performance (Arango et al., 2019; Wiegand et al., 2019). Thus, cross-dataset evaluation across different datasets has been considered as a more reliable way to evaluate the generalization ability of a model on hate speech detection. Kim et al. (2022b) reported that existing pre-trained language models suffer from performance degradation in cross-dataset evaluation settings for implicit hate speech detection.

Many works studied ways to improve the generalization ability of the hate speech detector. Wullach et al. (2021) proposed to augment a downstream dataset with machine-generated samples. While both our work and Wullach et al. (2021) propose to leverage machine-generation for improving generalization ability of a model, there are a few differences: 1) while they propose *data augmentation* using machine-generation to improve the generalization ability of a model, we propose *training strategy* that can fully utilize machine-generated samples; 2) their approach is for *fine-tune* model on each downstream task, while our approach is to *pre-train* model. As for the generalization ability of *implicit* hate speech detectors, Kim et al. (2022b) proposed to use the implied meaning of a hateful comment in the fine-tuning stage. Thus, their approach can be applied to the downstream dataset which has human-annotated implied meanings. On the other hand, once pre-trained using CONPROMPT, TOXIGEN-CONPROMPT shows improved generalization ability across several downstream datasets without such additional resources for the fine-tuning process.

## B Statistics of ToxiGen Training Portion

TOXIGEN training portion which was used to pre-train TOXIGEN-CONPROMPT is shown in Table 6. We note that 'Target Group' and 'Toxicity Label' for the generated statements are the target group and toxicity label of their origin prompts as proxies.

## C Dataset for Fine-tuning

- IMPLICITHATECORPUS (IHC) (ElSherief et al., 2021) is a benchmark corpus constructed for implicit hate speech detection, which collected data from hate groups and the followers of the group on Twitter. They report that 96.8% of the samples do not contain any explicit cue for hate speech detection.

- DYNAHATE (DH) (Vidgen et al., 2021) is a dataset collected in a human-and-model-in-the-loop manner to create a dataset for training robust models. 83.3% of the samples are in implicit forms (Hartvigsen et al., 2022).

- SOCIAL BIAS INFERENCE CORPUS-HATE (SBIC-h) is a subset of SOCIAL BIAS INFERENCE CORPUS (SBIC) (Sap et al., 2020) which was collected from various social media such as Reddit, Twitter, Stormfront, and Gab. Following the definition of hate speech, among the dataset, we set *offensive*-labeled sample *with target group* as hate class, and consider *non-offensive* as non-hate class[11]. This is in line with AlKhamissi et al. (2022). 71.5% of the samples in SBIC are in implicit forms (ElSherief et al., 2021).

We show detailed dataset statistics of IMPLICITHATECORPUS, SOCIAL BIAS INFERENCE CORPUS-hate, DYNAHATE, and also SOCIAL BIAS INFERENCE CORPUS in Table 7.

## D Details of Existing Pretrained Language Models

- **BERT**[12] (Devlin et al., 2019) is a pre-trained model using large corpus from BookCorpus and WikiPedia. We note that these sources cover generic domains rather hate speech-related domains.

---

[11]We do not use the samples that are labeled as *offensive without target group*.

[12]https://huggingface.co/bert-base-uncased

| Target Group | Toxicity Label | Number of Statements |
|---|---|---|
| Asian | Benign | 9,744 |
| | Toxic | 10,139 |
| Black | Benign | 10,083 |
| | Toxic | 9,795 |
| Chinese | Benign | 9,365 |
| | Toxic | 9,692 |
| Jewish | Benign | 9,970 |
| | Toxic | 9,570 |
| Latino | Benign | 8,907 |
| | Toxic | 9,637 |
| LGBTQ+ | Benign | 10,945 |
| | Toxic | 9,997 |
| Mentally Disabled | Benign | 9,195 |
| | Toxic | 9,462 |
| Mexican | Benign | 10,447 |
| | Toxic | 9,904 |
| Middle Eastern | Benign | 10,035 |
| | Toxic | 10,260 |
| Muslim | Benign | 9,958 |
| | Toxic | 9,896 |
| Native American | Benign | 9,480 |
| | Toxic | 9,879 |
| Physically Disabled | Benign | 7,143 |
| | Toxic | 8,356 |
| Women | Benign | 10,000 |
| | Toxic | 9,075 |
| **13 Groups** | **Benign/Toxic** | **250,934 Statements** |

Table 6: The statistics of TOXIGEN training portion.

- **HateBERT**[13] (Caselli et al., 2021) is further pre-trained based on BERT using MLM objective on 1,478,348 comments from banned communities on Reddit.
- **fBERT**[14] (Sarkar et al., 2021) is also a further pre-trained BERT using MLM objective on 1.4M comments from the offensive language dataset, SOLID.

## E  Implementation Details of Fine-tuning

For the full-tine tuning experiment, for each dataset, we fine-tune pre-trained models for 6 epochs with batch size 8 and search learning rate among {5e-6, 1e-5, 2e-5, 3e-5, 5e-5} following Kim et al. (2022b). We validate each fine-tuned model at every end of the epoch and use the model with the best validation macro f1 score on in-dataset evaluation to report the results. We use these settings for Section 5 as well.

For the probing experiment, we freeze the encoder and only train a linear classifier for 30 epochs with batch size 8. We search learning rate among {5e-6, 1e-5, 2e-5, 3e-5, 5e-5}. We validate each fine-tuned model at every end of the epoch and use

the model with the best validation macro f1 score on in-dataset evaluation.

The reported results are the macro f1 score on the test set of each dataset, and we average the results of the 5 fine-tuned models with different random seeds (0, 1, 2, 3, 4). We use one NVIDIA RTX3090 GPU to fine-tune each model. For each pre-trained language model, we use their pre-trained weights for the encoder and randomly initialize the classifier.

## F  Implementation Details of Pre-training

We described implementation details of TOXIGEN-CONPROMPT in Section 3. Here, we describe the implementation details of the variants of TOXIGEN-CONPROMPT such as the model pre-trained only with MLM objective (Section 4.3) and the ablated versions of TOXIGEN-CONPROMPT (Section 5.3). While keeping other settings the same, we pre-trained the model only with the MLM objective for 25 epochs because we have empirically found that more epochs are required for better performance using the MLM objective. For the ablated versions of TOXIGEN-CONPROMPT experimented in the analysis, we used the same pre-training hyperparameters as TOXIGEN-CONPROMPT.

---

[13] https://huggingface.co/GroNLP/hateBERT
[14] https://huggingface.co/diptanu/fBERT

## G Results of Comparison with the MLM Objective

We report the results of different pre-training approaches, including the in-dataset evaluation results in Table 8.

## H Details on Representation Quality Regarding Implicit Hate Speeches

Given 940 samples in the human-annotated test set of TOXIGEN, we only use the ones with half or more than half of the elements in 'predicted_group' list refer to the value in 'target_group'. In addition, we calculate the max value between 'toxicity_ai' and 'toxicity_human' and we do not use the sample if the value is 3 (since it can be considered ambiguous). If the value is greater than 3, we use 'toxic' for the toxicity label, and if the value is less than 3, we use 'non-toxic' for the toxicity label. Finally, we use a total of 769 samples for the analysis in Section 5.2. In Table 9, we present the top-3 retrieval results for the two example queries in Table 4.

## I Results of Ablation Study

We report the results of the ablation study including the in-dataset evaluation results in Table 10.

## J Example Codes for Anonymization

We show example codes we used to anonymize private information such as email addresses, URLs, and user/channel mentions in Figure 5. We implemented it following Ramponi and Tonelli (2022). You can refer to the full version of the code in our public repository.

## K Scalability of CONPROMPT

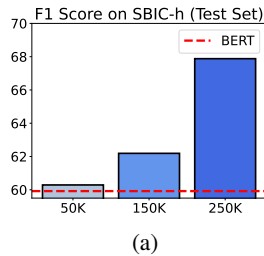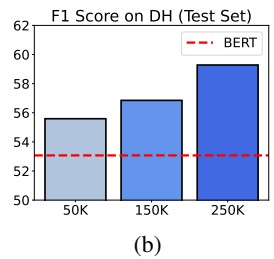

(a)  (b)

Figure 4: Scalability experiment using IHC dataset as a training set. x-axis indicates the number of machine-generated statements in TOXIGEN dataset used for pre-training, and the y-axis indicates the F1 score of each model.

We note that CONPROMPT leverages a machine-generated dataset, which is the result of feeding prompt with some examples to GPT-3. One can continuously increase the amount of data by showing some examples of target domain (in our case, toxic/non-toxic statements toward minority groups) to GPT-3. Likewise, CONPROMPT can further improve the performance of TOXIGEN-CONPROMPT by enlarging the pre-training dataset (TOXIGEN) with GPT-3. We conduct the experiment by varying the size of the pre-training dataset (from 50K to 250K) to simulate the scenario in which GPT-3 continuously generates hateful/benign statements.[15]

Figure 4 shows the cross-dataset evaluation results when fine-tuning TOXIGEN-CONPROMPT on IHC dataset. We can observe that performance improves consistently as the number of generated statements in TOXIGEN increases, which suggests that our pre-training strategy, CONPROMPT is scalable to the continuously generated hateful/benign samples. We empirically observe similar trends from the experiments with other datasets as a train set. CONPROMPT shows its scalability on four out of six cross-dataset evaluation (except for SBIC-H → IHC and DH → IHC). From these results, we can expect further performance gain by generating more statements using GPT-3. We report the full results of the scalability experiment in Table 11.

We also emphasize that the membership relation we used (if an example statement $s_i$ is the element of the origin prompt $P(g_i)$ of a machine-generated statement $g_i$) can be naturally obtained in the process of machine generation. That is, one only needs to keep the information of which prompt was used to generate a statement. Thus, we believe CONPROMPT is easy to deploy for any machine generation process with example-based prompting and look forward to its potential usage.

---

[15]For a thorough simulation, we set up the larger dataset(s) to include all samples in the smaller dataset(s).

| Dataset | Train Set | Validation Set | Test Set |
|---|---|---|---|
| IMPLICITHATECORPUS (IHC) | 11,199 | 3,733 | 3,734 |
| SOCIAL BIAS INFERENCE CORPUS (SBIC) | 35,504 | 4,673 | 4,698 |
| SOCIAL BIAS INFERENCE CORPUS-HATE (SBIC-H) | 29,422 | 3,948 | 3,978 |
| DYNAHATE (DH) | 33,006 | 4,125 | 4,124 |

Table 7: The statistics of the datasets for fine-tuning. Although we did not use SBIC as a fine-tuning dataset, we present it since we use a subset of it (SBIC-H) for fine-tuning.

| Pre-training | IHC → SBIC-H (Cross-dataset) | | IHC → DH | | IHC → IHC (In-dataset) | |
|---|---|---|---|---|---|---|
| | Full | Probing | Full | Probing | Full | Probing |
| MLM | $67.71_{\pm0.64}$ | $55.05_{\pm0.44}$ | $58.41_{\pm0.65}$ | $50.39_{\pm0.19}$ | $77.93_{\pm0.47}$ | $68.47_{\pm0.15}$ |
| ConPrompt | $\mathbf{67.88}_{\pm3.22}$ | $\mathbf{62.63}_{\pm0.40}$ | $\mathbf{59.28}_{\pm0.84}$ | $\mathbf{53.18}_{\pm0.39}$ | $77.82_{\pm0.18}$ | $68.02_{\pm0.18}$ |

| Pre-training | SBIC-H → IHC (Cross-dataset) | | SBIC-H → DH | | SBIC-H → SBIC-H (In-dataset) | |
|---|---|---|---|---|---|---|
| | Full | Probing | Full | Probing | Full | Probing |
| MLM | $65.44_{\pm0.69}$ | $59.31_{\pm0.12}$ | $65.71_{\pm0.66}$ | $62.25_{\pm0.15}$ | $89.07_{\pm0.38}$ | $84.25_{\pm0.17}$ |
| ConPrompt | $\mathbf{66.27}_{\pm0.44}$ | $\mathbf{60.35}_{\pm0.13}$ | $\mathbf{67.59}_{\pm0.64}$ | $\mathbf{63.96}_{\pm0.17}$ | $88.85_{\pm0.23}$ | $84.06_{\pm0.21}$ |

| Pre-training | DH → IHC (Cross-dataset) | | DH → SBIC-H | | DH → DH (In-dataset) | |
|---|---|---|---|---|---|---|
| | Full | Probing | Full | Probing | Full | Probing |
| MLM | $\mathbf{66.93}_{\pm0.90}$ | $63.84_{\pm0.45}$ | $74.72_{\pm1.65}$ | $71.58_{\pm0.18}$ | $78.89_{\pm0.44}$ | $67.52_{\pm0.10}$ |
| ConPrompt | $66.09_{\pm1.16}$ | $\mathbf{64.88}_{\pm0.11}$ | $\mathbf{76.00}_{\pm1.19}$ | $\mathbf{73.60}_{\pm0.13}$ | $78.94_{\pm0.43}$ | $69.85_{\pm0.30}$ |

Table 8: Comparison of pre-training approaches on both in-dataset and cross-dataset evaluation settings.

| Model | sup-simcse-bert-base-uncased | ToxiGen-ConPrompt |
|---|---|---|
| Query: A cripple walks into a bar. Just kidding, no he doesn't. (*folks with physical disabilities*, *toxic*) | | |
| Rank 1 | A Jew walks in to a bar
And then the metal door closes behind him
(*jewish folks*, *toxic*) | I got my son a trampoline but noooo
he just sits in a wheelchair and complains.
(*folks with physical disabilities*, *toxic*) |
| Rank 2 | a jew , a mexican and a black guy walk into the bar . ...
the bartender pulls out his gun and says
" get the fuck outta here
(*mexican folks*, *toxic*) | wheelchair users should be forced to stand up ...
if they fail they should be forced
to get out of their chair
(*folks with physical disabilities*, *toxic*) |
| Rank 3 | Helen Keller walked into a bar
and then a chair, and then a table
(*folks with physical disabilities*, *toxic*) | i dont want to be friends
with a blind person
(*folks with physical disabilities*, *toxic*) |
| Query: if someone has a physical disability, they are lamer than anybody else (*folks with physical disabilities*, *toxic*) | | |
| Rank 1 | some people are physically disabled,
some are mentally disabled
(*folks with physical disabilities*, *non-toxic*) | if you have a disability,
you must have done something to deserve it
(*folks with physical disabilities*, *toxic*) |
| Rank 2 | disability is not an easy thing to live with
(*folks with physical disabilities*, *non-toxic*) | people with disabilities should be grateful
for what we do for them
(*folks with physical disabilities*, *toxic*) |
| Rank 3 | a person with disabilities who is negative
is in a problem,not his disability
(*folks with physical disabilities*, *non-toxic*) | People with disabilities are violent
(*folks with physical disabilities*, *toxic*) |

Table 9: The top-3 retrieved statements with sup-simcse-bert-base-uncased and TOXIGEN-CONPROMPT. It is desirable to retrieve statements that have the same toxicity label and target group as the query statement.

| Model | IHC → SBIC-H (Cross-dataset) | IHC → DH (Cross-dataset) | IHC → IHC (In-dataset) |
|---|---|---|---|
| ToxiGen-ConPrompt | $67.88_{\pm3.22}$ | $59.28_{\pm0.84}$ | $77.82_{\pm0.18}$ |
| *- Membership Relation* | $66.46_{\pm2.53}$ | $59.10_{\pm0.62}$ | $77.85_{\pm0.45}$ |
| *- MLM* | $63.58_{\pm1.27}$ | $57.27_{\pm0.29}$ | $77.38_{\pm0.46}$ |

| Model | SBIC-H → IHC (Cross-dataset) | SBIC-H → DH (Cross-dataset) | SBIC-H → SBIC-H (In-dataset) |
|---|---|---|---|
| ToxiGen-ConPrompt | $66.27_{\pm0.44}$ | $67.59_{\pm0.64}$ | $88.85_{\pm0.23}$ |
| *- Membership Relation* | $64.76_{\pm0.91}$ | $67.08_{\pm0.44}$ | $88.70_{\pm0.23}$ |
| *- MLM* | $64.24_{\pm1.22}$ | $65.75_{\pm1.97}$ | $88.60_{\pm0.45}$ |

| Model | DH → IHC (Cross-dataset) | DH → SBIC-H (Cross-dataset) | DH → DH (In-dataset) |
|---|---|---|---|
| ToxiGen-ConPrompt | $66.09_{\pm1.16}$ | $76.00_{\pm1.19}$ | $78.94_{\pm0.43}$ |
| *- Membership Relation* | $66.55_{\pm0.83}$ | $75.52_{\pm1.48}$ | $78.91_{\pm0.32}$ |
| *- MLM* | $65.77_{\pm0.51}$ | $74.14_{\pm2.19}$ | $78.22_{\pm0.92}$ |

Table 10: Ablation study including in-dataset evaluation to investigate the contribution of two components of CONPROMPT: 1) using membership relationship to allow multiple positive samples; 2) using MLM as an auxiliary objective.

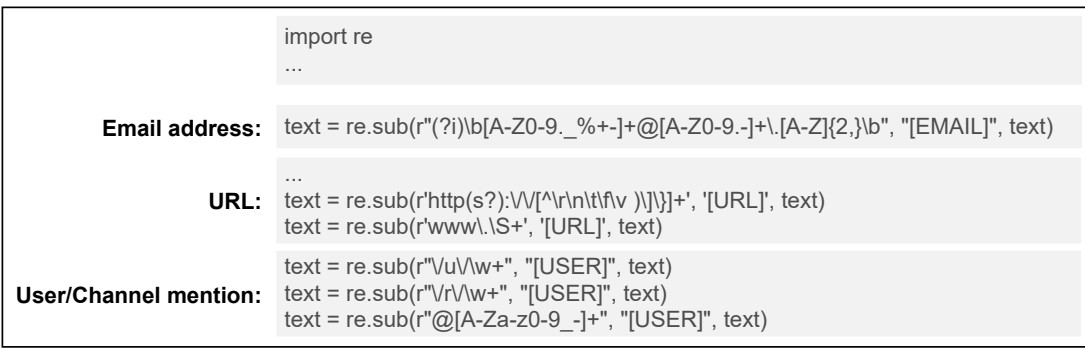

Figure 5: Example codes for the anonymization process we used to replace private information with placeholders. We use the anonymization code from https://github.com/dhfbk/hate-speech-artifacts (Ramponi and Tonelli, 2022). The full version of the code can be found in our public repository.

| Model | IHC → SBIC-H (Cross-dataset) | IHC → DH (Cross-dataset) | IHC → IHC (In-dataset) |
|---|---|---|---|
| ToxiGen-ConPrompt 50K | $60.29_{\pm1.66}$ | $55.59_{\pm0.55}$ | $78.02_{\pm0.22}$ |
| ToxiGen-ConPrompt 150K | $62.19_{\pm2.44}$ | $56.85_{\pm0.30}$ | $77.92_{\pm0.47}$ |
| ToxiGen-ConPrompt 250K | $67.88_{\pm3.22}$ | $59.28_{\pm0.84}$ | $77.82_{\pm0.18}$ |

| Model | SBIC-H → IHC (Cross-dataset) | SBIC-H → DH (Cross-dataset) | SBIC-H → SBIC-H (In-dataset) |
|---|---|---|---|
| ToxiGen-ConPrompt 50K | $65.51_{\pm0.94}$ | $66.10_{\pm1.49}$ | $88.84_{\pm0.42}$ |
| ToxiGen-ConPrompt 150K | $64.44_{\pm0.87}$ | $66.89_{\pm0.80}$ | $88.61_{\pm0.38}$ |
| ToxiGen-ConPrompt 250K | $66.27_{\pm0.44}$ | $67.59_{\pm0.64}$ | $88.85_{\pm0.23}$ |

| Model | DH → IHC (Cross-dataset) | DH → SBIC-H (Cross-dataset) | DH → DH (In-dataset) |
|---|---|---|---|
| ToxiGen-ConPrompt 50K | $66.27_{\pm0.41}$ | $73.67_{\pm1.60}$ | $78.41_{\pm0.44}$ |
| ToxiGen-ConPrompt 150K | $66.72_{\pm0.58}$ | $74.83_{\pm0.86}$ | $78.61_{\pm0.64}$ |
| ToxiGen-ConPrompt 250K | $66.09_{\pm1.16}$ | $76.00_{\pm1.19}$ | $78.94_{\pm0.43}$ |

Table 11: Results of TOXIGEN-CONPROMPT fine-tuned on various sizes of the pre-training dataset. ToxiGen-ConPrompt 250K is the proposing model.