# OpenReview forum: "ConPrompt: Pre-training a Language Model with Machine-Generated Data for Implicit Hate Speech Detection"
_EMNLP/2023/Conference — EMNLP 2023 Findings_

### Official Review · Reviewer_poeE · 2023-08-04

**Soundness:** 3

**Excitement:**

3: Ambivalent: It has merits (e.g., it reports state-of-the-art results, the idea is nice), but there are key weaknesses (e.g., it describes incremental work), and it can significantly benefit from another round of revision. However, I won't object to accepting it if my co-reviewers champion it.

**Paper Topic And Main Contributions:**

The main contributions of this paper include:
(1) The CONPROMT, a novel pre-training approach toward implicit hate speech detection by fully leveraging machine-generated data.
(2) TOXIGEN-CONPROMT, a pre-trained language model on the machine-generated implicit hate speech dataset, outperformed various other pre-trained hate speech languages.

**Reasons To Accept:**

The proposed pre-training approach and pre-trained model perform effectively in the task of detecting implicit hate speech. When compared to previous pre-trained models such as HateBERT and fBERT, the proposed TOXIGEN-CONPROMT outperforms on cross-dataset evaluation, indicating its capacity to generalize on this task. Furthermore, by learning desirable features in terms of implicit hate speech, the proposed pre-trained model is intended to decrease identity term bias.

**Reasons To Reject:**

It is highlighted that although taking advantage of the machine-generated model in terms of constructing datasets for training is a wise way to conduct experiments, procedures should be carefully considered by precise methods.

**Reproducibility:**

3: Could reproduce the results with some difficulty. The settings of parameters are underspecified or subjectively determined; the training/evaluation data are not widely available.

**Reviewer Confidence:**

5: Positive that my evaluation is correct. I read the paper very carefully and I am very familiar with related work.

---

> ### Author Rebuttal · Authors · 2023-08-29
>
> We deeply appreciate your constructive feedback. Here are our responses to your review.
>
> **[A carefully designed procedure of our work]**
>
>   We are delighted that you found our approach leveraging machine-generated data is a wise approach. It is true that with machine-generated data, we had to design our procedures by precise methods. We assess our work in terms of preciseness as follows:
>
>   *(Ethical Consideration)* It is crucial to care about the potential harms of using machine-generated data. We rigorously tackled the possible ethical issue regarding the machine-generated data. Before using the machine-generated data for pre-training, we inspected samples and observed that some private information was included in the machine-generated data. We anonymized such information following [a] to deal with the privacy issue (Line 272-280). We will describe our procedure of anonymization in a more detailed way in the camera-ready version for further preciseness.
>
>   *(Methodology)* We thoroughly describe our pre-training approach in Section 3. We first explain the objective based on the previous work [b] (Equation 2) and then present the modification of the objective (Equation 3). We justify the modification by providing the possible issue of Equation 2 regarding the false negatives within a mini-batch. Besides, we formulate our new way of selecting (multiple) positive samples using the membership relation (Line 239-244). Therefore, we believe that our methodology is presented in a precise way.
>
>   *(Validation)* We validate various aspects of our approach with precisely designed experiments. First, we conducted cross-dataset evaluation to validate the generalization ability of our approach (Section 4). Second, we measured the identity term bias of the models to validate the effect of our approach on reducing unintended behaviors (Section 5.1). Third, we compared the representation quality of our pre-trained model and the SimCSE model to validate the desirable representation quality of our model regarding implicit hate speech-related tasks (Section 5.2). Finally, we experimented with the ablated versions of our approach to investigate the contribution of each component and validate that our final approach is the best-performing version (Section 5.3). Through the thorough design of experiments, we conjecture that our validation is precise and appropriate.
>
>   *(Documentation)* For reproducibility, we reported the implementation details such as hyperparameters for both pre-training and fine-tuning in Appendix C and D due to the page limit. We will include the implementation details of our approach in the final version using an extra page.
>
>   We hope that our rebuttal helps you re-evaluate the soundness of our work.
>
>
> *[a] Ramponi, Alan, and Sara Tonelli. "Features or Spurious Artifacts? Data-centric Baselines for Fair and Robust Hate Speech Detection." Proceedings of the 2022 Conference of the North American Chapter of the Association for Computational Linguistics: Human Language Technologies. 2022.*
>
> *[b] Gao, Tianyu, Xingcheng Yao, and Danqi Chen. "SimCSE: Simple Contrastive Learning of Sentence Embeddings." Proceedings of the 2021 Conference on Empirical Methods in Natural Language Processing. 2021.*

---

### Official Review · Reviewer_tecQ · 2023-08-09

**Soundness:** 3

**Excitement:**

4: Strong: This paper deepens the understanding of some phenomenon or lowers the barriers to an existing research direction.

**Paper Topic And Main Contributions:**

Propose a pre-training approach, CONPROMPT for Implicit hate speech detection task

Propose TOXIGEN-CONPROMPT, a pre-trained BERT using TOXIGEN as a dataset



**Questions For The Authors:**

I think you should run some SOTA like : XLM-R.

ToxiGen-ConPrompt use ConPrompt as a dataset for training mlm, but i don't know it A Machine-generated Statement or A Human-curated Prompt or both of them.



**Reasons To Accept:**

A new way for generate Implicit hate speech and a new pretrain for this task.

**Reasons To Reject:**

The dataset for training ToxiGen-ConPrompt is machine-generated data, so this is not good data for pretrain-model such as bert.



**Reproducibility:**

5: Could easily reproduce the results.

**Reviewer Confidence:**

2: Willing to defend my evaluation, but it is fairly likely that I missed some details, didn't understand some central points, or can't be sure about the novelty of the work.

---

> ### Author Rebuttal · Authors · 2023-08-29
>
> We appreciate your constructive questions and feedback. Here are our responses to your review.
>
> **[Using machine-generated data for pre-training a model]**
>
>   We respectfully disagree with your opinion that machine-generated data is not good data for pre-training a model.
>
>   First, the machine-generated data (i.e., ToxiGen) we used is of high quality. The authors of ToxiGen conducted various human evaluations and confirmed its high quality. For example, about 90.5% of the machine-generated samples were considered to be written by humans in the human evaluation.
>
>   Second, there are several works showing the effectiveness of using machine-generated data for training a model. For example, [a] generated a large-sized commonsense knowledge graph using GPT-3 and trained a commonsense model using the machine-generated data. They empirically showed the effectiveness of using machine-generated data over human-generated one.
>
>   Therefore, we believe that leveraging machine-generated data for pre-training a model is a promising direction.
>
>
> **[Experiment with XLM-R]**
>
>   We respectfully disagree with your opinion that we should run XLM-R as a SOTA model. We remark that we only use the pre-trained models that are based on bert-base-uncased on purpose.
>
>   First, we respectfully disagree with your idea that XLM-R is a SOTA model for implicit hate speech detection. Our baselines such as HateBERT and fBERT were pre-trained on hate speech-related corpus and showed SOTA performance on various hate speech detection datasets. However, XLM-R was pre-trained on general corpus and is not specialized in hate speech detection.
>
>   Second, by using only BERT-based models (i.e., the models had the same starting point), various pre-training strategies can be fairly compared (Line 327-330). Through the fair comparison, we were able to successfully validate the effectiveness of our pre-training approach and our pre-trained language model.
>
>   As you mentioned, RoBERTa-based models like XLM-R can be employed as alternatives replacing BERT-based models, which is also an intriguing direction. We will leave this as a future direction.
>
>
> **[Training resources: machine-generated statements and human-curated prompts]**
>
>   ToxiGen-ConPrompt used ToxiGen as a dataset by leveraging ConPrompt as a pre-training approach with MLM as an auxiliary objective. For ConPrompt objectives, we utilize both the machine-generated statements and the human-curated prompts (Line 88-91, 210-217, Equation 3, and Figure 2). For MLM objectives, we use machine-generated statements (Line 255-256). We will clarify this in the camera-ready version.
>
>
> *[a] West, Peter, et al. "Symbolic Knowledge Distillation: from General Language Models to Commonsense Models." Proceedings of the 2022 Conference of the North American Chapter of the Association for Computational Linguistics: Human Language Technologies. 2022.*

---

### Official Review · Reviewer_jhCf · 2023-08-09

**Soundness:** 3

**Excitement:**

4: Strong: This paper deepens the understanding of some phenomenon or lowers the barriers to an existing research direction.

**Paper Topic And Main Contributions:**

- The authors proposed a pre-training strategy, named ConPrompt, in order to fully leverage machine-generated data. This strategy utilizes examples from a prompt as positive samples for contrastive learning. By applying ConPrompt to the machine-generated implicit dataset ToxiGen and modifying “bert-base-uncased”, the authors achieved ToxiGen-ConPrompt. This model out-performed various pre-trained hate speech language models, including HateBERT and fBERT, in a cross-dataset evaluation, showcasing its generalization ability in implicit hate speech detection. Besides, ToxiGen-ConPrompt proved to be effective in reducing identity term bias.

**Questions For The Authors:**

- Question A: The paper presented very promising results when evaluated on cross-dataset performance. However, the performance on the in-dataset seems to have reached a threshold even though it was trained on ToxiGen, a high-quality dataset that improved cross-dataset performance. Could the authors provide further clarification on this limitation?

**Reasons To Accept:**

- Based on thorough experimentation across various implicit hate speech cross-datasets, ToxiGen-ConPrompt demonstrated a notably superior capacity for generalization compared to other pre-trained models.
- ToxiGen-ConPrompt effectively addresses the issue of identity term bias, showcasing its dual capability to enhance model generalizability while mitigating unintended biases.

**Reasons To Reject:**

- Despite the superior results of ToxiGen-ConPrompt compared to HateBERT and fBERT in the cross-dataset evaluation, the performance differences within the in-dataset evaluation are not significantly pronounced.
- I agree that this paper proposes a method that emphasizes pre-training the language model rather than focusing on the fine-tuning phase, as done in [1] and [2]. However, I believe this paper would be improved by describing the differences in data characteristics between ToxiGen corpus and the machine-generation data used in [1] and [2].

References:
- [1] Paul Röttger, Bertie Vidgen, Dong Nguyen, Zeerak Waseem, Helen Margetts, and Janet Pierrehumbert. 2021. HateCheck: Functional tests for hate speech detection models. In Proceedings of the 59th Annual Meeting of the Association for Computational Linguistics and the 11th International Joint Conference on Natural Language Processing (Volume 1: Long Papers), pages 41–58.
- [2] Youngwook Kim, Shinwoo Park, and Yo-Sub Han. 2022. Generalizable implicit hate speech detection using contrastive learning. In Proceedings of the 29th International Conference on Computational Linguistics, pages 6667–6679.


**Reproducibility:**

4: Could mostly reproduce the results, but there may be some variation because of sample variance or minor variations in their interpretation of the protocol or method.

**Reviewer Confidence:**

3: Pretty sure, but there's a chance I missed something. Although I have a good feel for this area in general, I did not carefully check the paper's details, e.g., the math, experimental design, or novelty.

---

> ### Author Rebuttal · Authors · 2023-08-29
>
> We appreciate your helpful questions and comments. Here are our responses to your review.
>
>
> **[Why we focused on cross-dataset evaluation]**
>
>   We respectfully disagree with you regarding the limitation of our approach on in-dataset evaluation.
>
>   While ours showed comparable performance on in-dataset evaluation, our main research direction is to improve the performance for the cross-dataset evaluation. This is because the in-dataset evaluation is known to have several limits in hate speech detection tasks due to unintended biases such as identity term bias. Thus, the cross-dataset evaluation is widely utilized in hate speech detection tasks to assess the generalization ability of a model in the literature (Line 292-300, 758-774). This is why we focused on cross-dataset evaluation, where ours showed superior performance.
>
>   Therefore, we believe that our experimental results strongly support the superiority of our approach in hate speech detection, and it is not a limitation to have a comparative performance for in-dataset evaluation. We will elaborate on this using an extra page in the camera-ready and hope that our rebuttal can help you re-evaluate the soundness of our work.
>
>
> **[Differences in data characteristics with other works]**
>
>   Before starting, we presume that you meant “human”-generation data (instead of “machine”-generated data in your review) used in [1] and [2]. Two main differences in data characteristics are as follows:
>
>   1. The use of the machine generation differs. While ToxiGen was constructed using machine (e.g., GPT-3) generation, the datasets in [1] and [2] were not. In HateCheck [1], they used templates and replaced the placeholders with pre-defined group identifiers and slurs. The datasets in [2] were mostly collected from social media and human-annotated.
>
>   2. The size and the aim of each dataset is different. ToxiGen is a large-scale dataset with over 250k samples, and we used the dataset for pre-training. However, HateCheck [1] is a small dataset with 3,728 samples, and it was constructed for the evaluation of a model. In [2], they used three datasets with 18k to 44k samples and used the dataset for fine-tuning.
>
>
>   Hopefully, our rebuttal clarifies your concerns about our contribution in terms of its correctness.

---

### Official Review · Reviewer_hY1k · 2023-08-12

**Soundness:** 4

**Excitement:**

4: Strong: This paper deepens the understanding of some phenomenon or lowers the barriers to an existing research direction.

**Paper Topic And Main Contributions:**

This paper introduces ToxiGen-ConPrompt, a pre-trained BERT model designed for implicit hate speech detection. It leverages the construction strategy of the ToxiGen dataset to generate positive samples, which are then utilized in a contrastive learning approach for the task of implicit hate speech detection. The ToxiGen dataset is crafted through ChatGPT, utilizing manually designed prompts for generating an implicit hate speech detection dataset. This paper employs the original prompt of a sample as a positive example for that sample. Subsequently, an adapted version of simCSE is utilized to enhance the closeness of representations for similar samples. Finally  they use cross-dataset evaluation across  different  implicit datasets to showcase the generalizability of their proposed method.

**Reasons To Accept:**

The paper exhibits excellent writing and a well-organized structure. It encompasses a variety of experiments, including those conducted on the representation space, to demonstrate the superiority of their modified contrastive objective in comparison to its previous version.
They ingeniously employed a creative approach to generate positive samples for contrastive learning. Additionally, they recognized the potential presence of other similar samples within the batch and introduced an adapted version of the contrastive objective function to account for this possibility.

**Reasons To Reject:**


The generalizability of their method is assessed in comparison to other existing PLMs (fBERT, HateBERT, BERT), revealing the relatively superior performance of ToxiGen-ConPromt. However, it's worth noting that these mentioned PLMs were not specifically designed to address the issue of degradation on cross-dataset evaluation. It would be reasonable to compare their method with approaches that are explicitly design to tackle this problem, such as the two methods mentioned in the related work appendix specially  "Generalizable Implicit Hate Speech Detection using Contrastive Learning" (kim et al. 2022).

**Reproducibility:**

5: Could easily reproduce the results.

**Reviewer Confidence:**

4: Quite sure. I tried to check the important points carefully. It's unlikely, though conceivable, that I missed something that should affect my ratings.

**Typos Grammar Style And Presentation Improvements:**

Line 388, if I understand correctly, you mean that "ConPrompt" is your pre-training approach, so you should insert a comma after "ConPrompt."

---

> ### Author Rebuttal · Authors · 2023-08-29
>
> We deeply appreciate your constructive feedback. Here are our responses to your review.
>
> **[Comparison with other methods ([a] and [b]) addressing the generalization issue]**
>
>   We believe that it is fairer to compare our pre-trained model with other pre-trained models instead of [a] and [b]. We remark that while [a] and [b] are designed for fine-tuning process (i.e., they require the corresponding downstream dataset), our approach (ConPrompt) is for pre-training process (i.e., ours does not require such a downstream dataset). This is why we compared our approach with other PLMs rather than the fine-tuning processes (i.e., [a] and [b]).
>
>   Nonetheless, we agree that it is important to discuss the relationship between our pre-training approach and other fine-tuning methods including [a] and [b]. Particularly, our approach and other methods can be used together for further improvement. For example, one can use our approach in the pre-training step, and then apply the method of [a] or [b] in the fine-tuning process. We will elaborate on this in the camera-ready version.
>
> **[Suggestion regarding the typo]**
>
>   We appreciate your suggestion regarding Line 388 and will revise it considering your suggestion in the camera-ready version. (i.e., the revised sentence would be “... we compare our pre-training approach–ConPrompt–with the MLM objective.”)
>
>
> *[a] Kim, Youngwook, Shinwoo Park, and Yo-Sub Han. "Generalizable implicit hate speech detection using contrastive learning." Proceedings of the 29th International Conference on Computational Linguistics. 2022.*
>
> *[b] Wullach, Tomer, Amir Adler, and Einat Minkov. "Fight Fire with Fire: Fine-tuning Hate Detectors using Large Samples of Generated Hate Speech." Findings of the Association for Computational Linguistics: EMNLP 2021. 2021.*

---

### Meta-Review · Area_Chair_UdtG · 2023-09-19

**Recommendation:** 2

**Metareview:**

The paper primarily introduces a pre-trained BERT model tailored for implicit hate speech detection. The paper introduces ToxiGen dataset, which is built with ChatGPT, to generate positive samples. The paper also introduces a contrastive learning appraoch called ConPrompt. This contrastive learning approach similar to simCSE.

One of the key claims in the paper is the cross-data set performance. This is a nice feature as exposure to identity terms in one dataset can cause the model to be overly confident on that data, and not transfer well to unseen task settings. However, while the model does exhibit performance increases in this transfer setting. It is not exactly clear whether this is an issue with the model or the dataset. Examining table 1, Large benefits are provided when trained on IHC, but these are much less pronounced with SBIC-H and DH. There was no discernable difference in the in-dataset setting.

The paper received mixed but positive reviews from the reviewers. The reviewers have concerns about the use of machine-generated data. Additionally, ethical concerns regarding privacy and potential misuse were raised by the ethics meta reviewer. The use of machine-generated data, while innovative, necessitates a thorough discussion on its implications, especially in a sensitive domain like hate speech detection.

---

### Decision · Program_Chairs · 2023-10-07

**Decision:**

Accept-Findings

**Comment:**

The paper primarily introduces a pre-trained BERT model tailored for implicit hate speech detection. The paper introduces ToxiGen dataset, which is built with ChatGPT, to generate positive samples. The paper also introduces a contrastive learning appraoch called ConPrompt. This contrastive learning approach similar to simCSE.

One of the key claims in the paper is the cross-data set performance. This is a nice feature as exposure to identity terms in one dataset can cause the model to be overly confident on that data, and not transfer well to unseen task settings. However, while the model does exhibit performance increases in this transfer setting. It is not exactly clear whether this is an issue with the model or the dataset. Examining table 1, Large benefits are provided when trained on IHC, but these are much less pronounced with SBIC-H and DH. There was no discernable difference in the in-dataset setting.

The paper received mixed but positive reviews from the reviewers. The reviewers have concerns about the use of machine-generated data. Additionally, ethical concerns regarding privacy and potential misuse were raised by the ethics meta reviewer. The use of machine-generated data, while innovative, necessitates a thorough discussion on its implications, especially in a sensitive domain like hate speech detection.